# Modeling the Relationship between Input Distributions and Learning Trajectories with the Tolerance Principle

**Jordan Kodner**
Stony Brook University
Department of Linguistics
Institute for Advanced Computational Science
Stony Brook, NY, USA
`Jordan.Kodner@stonybrook.edu`

## Abstract

Child language learners develop with remarkable uniformity, both in their learning trajectories and ultimate outcomes, despite major differences in their learning environments. In this paper, we explore the role that the frequencies and distributions of irregular lexical items in the input plays in driving learning trajectories. I conclude that while the Tolerance Principle, a type-based model of productivity learning, accounts for *inter-learner uniformity*, it also interacts with input distributions to drive *cross-pattern variation* in learning trajectories.

## 1 Introduction

One of the most striking characteristics of child language acquisition is its uniformity (Labov, 1972). Children in the same speech community acquire the same grammars despite the lexical variation in each child's individual input: a recent quantitative study of child-directed speech (CDS) finds Jaccard similarities of only 0.25-0.37 between individual portions of the Providence Corpus (Richter, 2021), not much higher than the lexical similarity between CDS and adult genres (Kodner, 2019). Thus, to explain uniformity of outcomes, grammar learning must not depend primarily on lexical identity but on more general patterns in the learner's input.

Learners not only acquire the essentially same grammars but acquire them following similar trajectories. For example, English learners consistently acquire the verbal *-s* and *-ing* before the past *-ed* (Brown, 1973), the last of which shows a *u*-shaped developmental regression (Ervin and Miller, 1963; Pinker and Prince, 1988). Individuals may show relative delays correlating to estimated working vocabulary size (Fenson et al., 1994, ch. 5-6), but variability is otherwise limited. However, while individuals learning the same pattern show uniformity, expected learning paths vary across patterns. Among English learners, for example, *-ing* does not show *u-shaped* learning, unlike *-ed*. Children

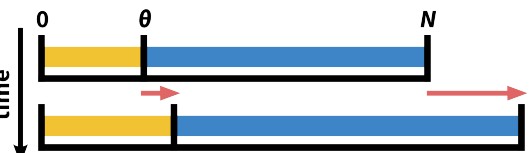

Figure 1: Visualizing the Tolerance Principle on a number line. $e$ falls in the range $[0, N]$. If it lies below $\theta$ (gold), then the learner should acquire the pattern and memorize the exceptions. If $e$ lies above $\theta$ (blue), the learner should resort to memorization instead. The number line extends as the learner's vocabulary grows.

learning Spanish verb stem alternations also show *u*-shaped learning, but they begin to over-regularize a year earlier than English past tense learners (Clahsen et al., 2002). One potential reason for this, differences in patterns' distributions in the input, is investigated here.

This paper introduces a quantitative means of assessing the role that the distribution of linguistic patterns in learner input plays in shaping learning trajectories and variation even prior to the grammar and individual cognitive factors. Adopting the Tolerance Principle (TP; Yang, 2016) as a type-based model of productivity learning, we find that the type-frequency and (indirectly) token frequency of exceptions to linguistic patterns have a dramatic effect on the expected learning trajectories across patterns while also quantifying expected uniformity across individuals within a given pattern.

## 2 The Learning Model

The *Tolerance Principle* (TP; Yang, 2016) is a cognitively motivated type-based learning model which casts generalization in terms of productivity in the face of exceptions. The model has gained support in recent years through its successful application to problems in syntax and semantics (e.g., Yang, 2016; Irani, 2019; Lee and Kodner, 2020), morphology (e.g., Yang, 2016; Kodner, 2020; Björnsdóttir, 2021; Belth et al., 2021),

and phonology (e.g., Yang, 2016; Sneller et al., 2019; Kodner and Richter, 2020; Richter, 2021). It has increasingly received backing from a range of psycholinguistic experiments (Schuler, 2017; Koulaguina and Shi, 2019; Emond and Shi, 2020). It is adopted here because it makes categorical and auditable predictions about productivity and thus provides a clear means for investigating and the relationship between distributions in the input and the dynamics of learning.

The TP serves as a decision procedure for the learner. Once the learner hypothesizes a generalization in the grammar, it establishes the threshold $\theta_N$ at which it becomes more economical in terms of lexical access time to accept the hypothesis and exceptions rather than to just memorize items individually. (1) formalizes the TP. The *tolerance threshold* $\theta_N$ is defined as the number of known types that a generalization should apply to divided by its natural logarithm.[1]

(1) **Tolerance Principle** (Yang, 2016, p. 8):
If $R$ is a productive rule applicable to $N$ candidates, then the following relation holds between $N$ and $e$, the number of exceptions that could but do not follow $R$:

$$e \leq \theta_N \text{ where } \theta_N := \frac{N}{\ln N}$$

The derivation of the TP acknowledges that items in the input follow long-tailed Zipfian frequency distributions (Zipf, 1949) in which few items are well-attested and others are rarely attested in the input. Zipfian and other long-tailed distributions are quite common throughout language and are very prominent in lexical and inflectional frequencies (e.g., Miller, 1957; Jelinek, 1997; Baroni, 2005; Chan, 2008; Yang, 2013; Lignos and Yang, 2018)

Figure 1 provides a visualization of the Tolerance Principle over individual development. Crucially, $N$ depends on a learner's current working vocabulary and is not a comment on the language's vocabulary in general. An individual learner's $N$ and $e$ increase as they learn more vocabulary, and a pattern may fall in and out of productivity.

## 3 Input Distributions driving Trajectories

This section uses the Tolerance Principle to calculate likely learning trajectories and variability in learning trajectories given distributions of regular and irregular forms in the input, and it discusses the impact that input distributions have on learning paths. It presents two illustrative examples and a case study from English past tense learning. For clarity, $N_{tgt}$ and $e_{tgt}$ are used here to represent the expected mature learner state, since $N$ and $e$ properly represent speaker-internal quantities and are not a description of the target language.

### 3.1 Calculating Trajectories with the TP

In the first illustrative example, $N_{tgt} = 82$ and $e_{tgt} = 32$. This pattern should not be productive for a mature speaker ($e_{tgt} > \theta_{Ntgt} = 18.6$), but learners may pass through a period of over-generalization if their $N$ and $e$ support it at some point during development. To help with conceptualizing these developments, I introduce a visualization called a Tolerance Principle state space for this system in Figure 2. The $x$-axis indicates the number of regular forms that an individual has learned so far ($N - e$), and the $y$-axis indicates the number of irregular forms learned so far. Color indicates whether or not a learner at ($N - e, e$) should learn a productive generalization. These are the two "zones" in the state space. The bottom left corner, $N = 0$, indicates the initial state for all learners, and the top right corner ($N = N_{tgt}$), indicates the mature state. In this example, the final state is in the non-productive zone.[2]

As learners mature, they "move" through the state space along some path from the bottom left to top right. The paths that individuals take are a function of the order in which they personally acquired regular and irregular items. Learners may pass in and out of the productive zone as they develop. In this example, a learner who passes temporarily through the productive zone may produce over-generalization errors, one source of *u*-shaped learning.

Not all paths through the state space are equally likely. It would be strange, for example, if a learner acquired all the irregular items before any of the regular items, or vice-versa. One could ask, for a learner who knows a given $N$, what is the likelihood that $e$ of those are irregulars? Or equivalently in the state space, what is the likelihood that a learner should pass through a given point ($N - e, e$)? This can be modeled probabilistically

---

[1]See Yang (2016, pp. 10, 144) for the full mathematical derivation. $\theta_N$ approximates the $N$th harmonic number

[2]The TP breaks down for very small $N$. This area is placed in the non-productive zone by convention.

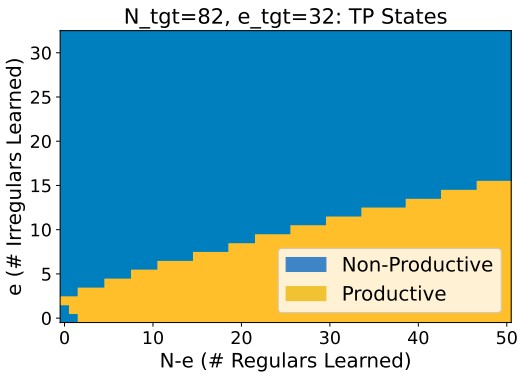

Figure 2: Tolerance Principle state space indicating productivity for every $(N - e, e)$ pair that a learner may pass through during vocabulary learning. $N_{tgt}$=82, $e_{tgt}$=32.

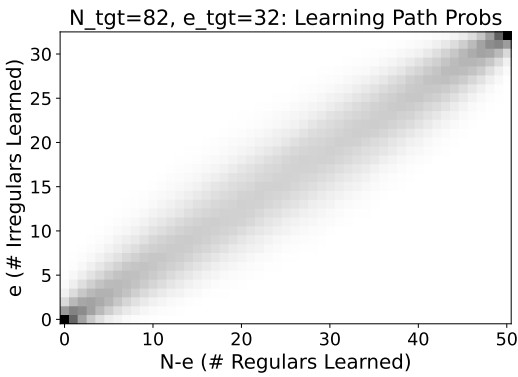

Figure 3: Likelihood of $(N - e, e)$ for each $N$. Darker indicates more likely path through the Fig. 2 TP space.

as a function of the relative token frequencies of the items. If irregulars are distributed uniformly throughout the distribution of types, path likelihood is well-approximated by a central hypergeometric distribution calculated for each $N$. Diagonals from top left to bottom right are "lines of constant $N$." Figure 3 visualizes this, with darker colors indicating more likely ratios of regulars and irregulars for a given $N$.

It is now possible to calculate the probability of falling in the productive and non-productive zones for each vocabulary size by summing over lines of constant $N$. The results, visualized in Figure 4 can be interpreted as the probability that a learner will generalize at each vocabulary size. Correlated with vocabulary size estimates by age, this can predict developmental trajectories. In this example, learners are will pass through a phase of early overgeneralization. This falls rapidly such that only about half should overgeneralize at $N = 15$. There is still a non-zero chance of over-generalizing be-

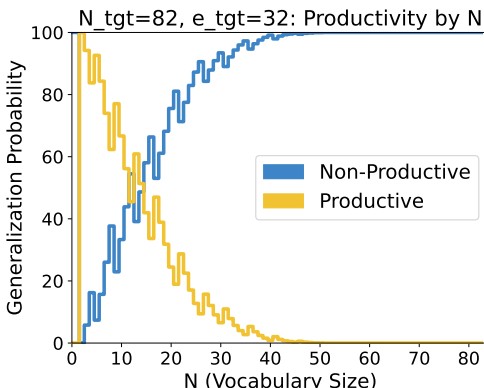

Figure 4: Likelihood of generalization and non-generalization by vocabulary size for Figs. 2-3.

fore $N = 45$, but after that point, all learners converge on adult-like non-productivity.

Note that productivity is driven entirely by the relative number of lexical items that follow or disobey the learner's hypothesized generalization and not the presence or absence of any individual lexical items. Learner outcomes are instead driven directly by the type frequency of patterns and the TP. Token frequencies play an indirect but crucial role as well. They determine the likely relative order that regular and irregular items are learned. The second illustration demonstrates this.

### 3.2 Effect of Irregular Token Frequency

This illustrative example examines the effect of irregular token frequency on learning trajectories by adopting a more realistic Zipfian input distribution.[3] The pattern $N_{tgt} = 90$, $e_{tgt} = 18$ should be acquired productively ($N_{tgt}$ is in the productive zone of the state space visualized in Figure 5).

The 90 items are assumed to be distributed according to a Zipfian distribution. This should bow the most likely path through the state space, potentially pushing it into our out of the productive zone.[4] For example, if irregulars tend to fall on the frequent end of the distributions, these will tend to be heard, and therefore acquired earlier. This should bow the likely path upward and deeper into the non-productive zone. Three irregular distribu-

---

[3] Irregulars are often clustered in the high-frequency range (e.g., English past tense), but this is not universal. Other irregulars are more uniformly distributed in CDS (e.g., English plurals, Spanish verbs (Fratini et al., 2014)).

[4] Directly calculating each $(N - e, e)$ probability is intractable if every item has its own frequency. Wallenius' noncentral hypergeometric distr. allows class but not item weighting and was found to be a poor approximation. Thus, probabilities were calculated by simulating 100,000 trials.

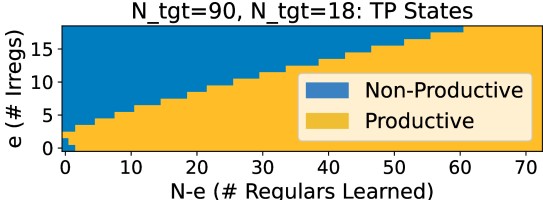

Figure 5: TP state space for $N_{tgt} = 90$, $e_{tgt} = 18$. This pattern should be acquired productively.

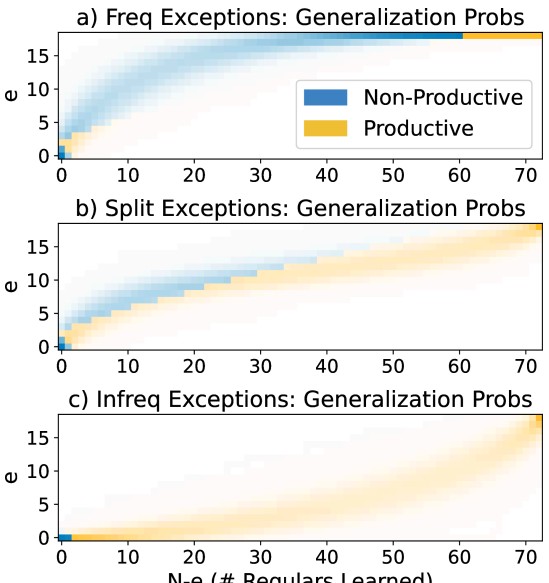

Figure 6: Likelihood of $(N - e, e)$ for each $N$ and a) top-heavy, b) split, c) bottom-heavy $e$ distributions.

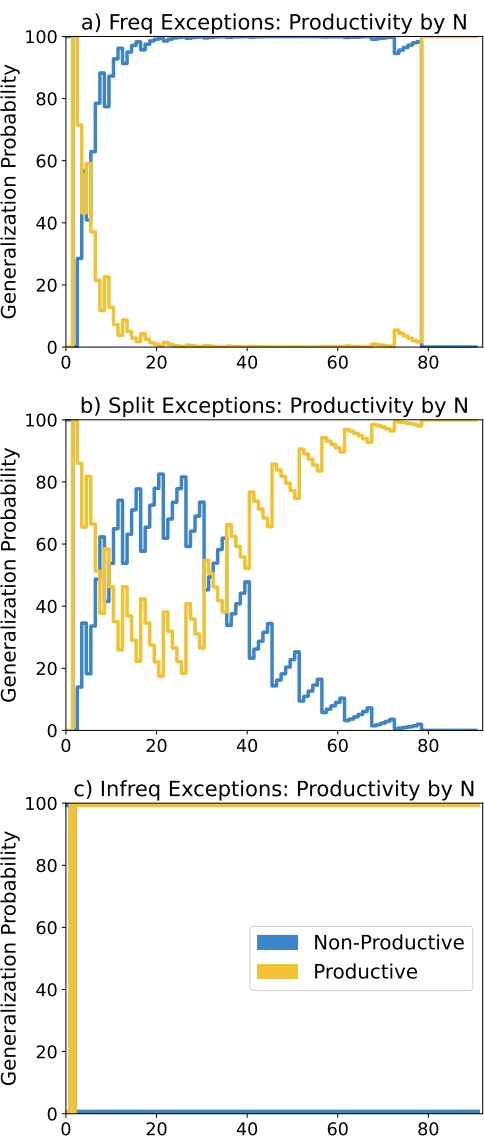

Figure 7: Likelihood of generalization and non-generalization by vocabulary size given Fig. 6.

tions are tested: They are **a)** the 18 most frequent items (the head of the Zipfian curve), **b)** the 9 most frequent and 9 least frequent items, and **c)** the 18 least frequent items. They are visualized in Figure 6 for three distributions of irregulars:

Even though the type distribution is the same in each case, the expected learning trajectories differ dramatically (Fig. 7). In the top-heavy case, nearly no learners are expected to be productive between $N = 20$ and $N = 80$, then everyone rapidly achieves productivity. In the bottom-heavy all learners achieve productivity as soon as they hypothesize the generalization. The split case predicts transient variation where all early learners are essentially adult-like, but many temporarily abandon productivity before relearning it later. This is because the likely path through the TP state space skirts the tolerance threshold, so slight variation in each individual's $e$ predicts a large categorical difference in the grammar.

## 3.3 Application to English Past Tense

This section applies the methods described thus far to real data: English past tense items extracted with frequencies from the CHILDES database (MacWhinney, 2000). Two expected learning paths were calculated: the default past *-ed* ($N = 1328$, $e = 98$ in this data) and the relatively common *sing-sang, ring-rung* sub-pattern ($N$=26, $e$=8). English learning children consistently acquire productive *-ed* around age three (Berko, 1958; Marcus et al., 1992). In contrast, the *sing-sang* pattern is not productive, though there is some transient variation (Berko, 1958; Xu and Pinker, 1995; Yang, 2016). This is because it has many exceptions (e.g., *sting-stung, bring-brought*).

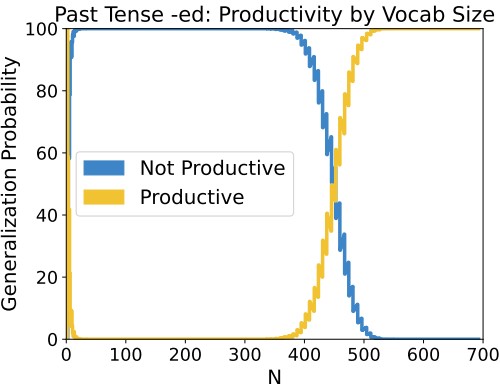

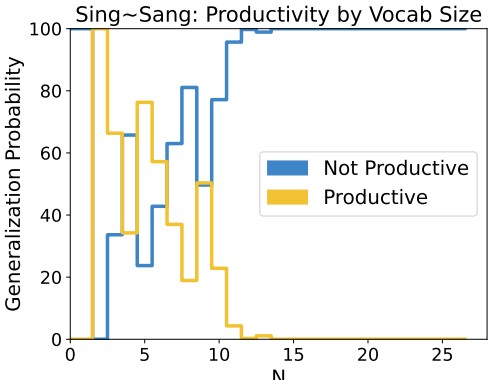

Figure 8: Generalization probability by vocabulary size for English past tense *-ed* and *sing-sang*. *-ed* was calculated on all data but trimmed to $N$=700 for visualization.

Figure 8 visualizes the results. Learners are predicted to show great uniformity in the acquisition of *-ed*. They consistently acquire the rule when they know 400-500 verbs. This qualitative uniformity is consistent with known developmental facts, but it is not immediately clear whether these particular numbers line up with the empirical evidence. Estimates of vocabulary size by age vary by method, but Marcus et al. (1992, ch. 5) report that Sarah and Adam from the Brown Corpus have produced 300-350 unique verbs by age three, but productive vocabulary underestimates working knowledge (Fenson et al., 1994, ch. 5-6), which is what is being modeled here.

The predictions for *sing-sang* is quite a bit different. There is significant variability when vocabulary size is small, but learners uniformly decide on non-productivity by around $N$=12. This appears to be consistent with wug-test results for children. In the original Berko (1958) study, only three of 86 pre-schoolers produce an *-ang(ed)* past form for stimuli *gling*+PAST or *bing*+PAST, suggesting low variability and low-productivity in that age group.[5]

---

[5]Adults and children seem to approach the wug test differently (Schütze, 2005), with many adults treating it as an analogy game (Derwing and Baker, 1977). Adults can be prompted to analogize the *sing-sang* pattern Berko (1958)

## 4 Discussion

This paper presents a means of modeling expected learning trajectories for productivity using the Tolerance Principle. As a type-based model of productivity learning, the TP only relies directly on the type attestation of regular and irregular items in the input. Since the grammar which is learned only depends on which side of the tolerance threshold the number of irregulars falls and not the lexical identity of the items or their exact number, it explains the general uniformity of outcomes observed across individual learners.

The TP was derived assuming that learners expect long-tailed frequency distributions in their input, and it provides an indirect role for token-frequency in learning. Higher frequency items are more likely to be attested early and learned early. Thus *while the type distribution of irregulars governs the ultimate learning outcome, their token distribution drives the learning trajectory*: the vocabulary size at which the adult-like grammar is settled on, the likelihood of over-regularization, and the degree of variability among individual learners.

One advantage of the TP for the purposes of this type of modeling is that it makes clear binary predictions about productivity. This study provides a novel means for making concrete predictions about the learning paths predicted by the TP. It remains to be seen how well these predictions fit the empirical data in a wider range of case studies. Another open question is whether other generalization models would make similar or different predictions, and if so, which best fit the empirical data.

The distribution of irregulars in the input can be measured empirically from corpora of child-directed speech since it is a property of the lexicon and of discourse concerns. The input has a clear effect on the path of learning even prior to adopting specific assumptions about the underlying grammar that children acquire. This suggests quantitatively re-evaluating the input as a way forward for explaining cross-linguistic differences in child language development as a complement to cross-linguistic theoretical and experimental work.

## Acknowledgements

I am grateful to Mark Aronoff, Caleb Belth, Kenneth Hanson, Jeff Heinz, Sarah Payne, Charles Yang, and an audience at Stony Brook University for feedback they provided on drafts of this work.

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
