# OpenReview forum: "Modeling the Relationship between Input Distributions and Learning Trajectories with the Tolerance Principle"
_aclweb.org/ACL/2022/Workshop/CMCL — CMCL 2022_

### Official Review · Reviewer_Skum · 2022-03-24
**Quantitatively assessing impact of child language input distributions on developmental learning trajectories**

**Rating:** 7
**Confidence:** 4

**Review:**

This paper models the relationship of child language input distributions on learning trajectories. This work is based on a cognitively motivated type-based learning model known as the Tolerance Principle that estimates a threshold of minimum exceptions to a linguistic pattern that are allowed for the language learner to productively use (and presumably influence decision-making and learning) a generalization in the grammar (in contrast to just memorizing individual items). The TP model has previously been used to account for inter-learner uniformity or how children acquire the same grammatical rules despite different learning input/environments (e.g., lexical variation).

The authors show that the TP model accurately predicts how the type distribution of irregulars governs the overall learning outcome in line with previous work showing it can account for inter-learning uniformity. However, the novelty of this paper is in using the TP model to quantitatively assess how input distributions may drive variation in developmental learning trajectories. Specifically, how the token distribution (relative token frequencies or the order of acquisition of regular and irregular items) can lead to different learning trajectories.
The authors show this capability through an example using three different distributions of irregulars and their dramatic impact on the learning trajectory. They then show an application to real-world data mainly showing how this model can be used to quantify (visualize) the learning trajectories of the productive English past tense (the default past -ed) and the unproductive sing-sang sub-pattern drawing on CHILDES database.

This has potential application towards quantitatively assessing how variation in input distributions accounts for observed cross-linguistical developmental learning trajectories. It would be great if the authors could give a specific example of this if the space allows for it as it would give the paper more breadth. I would also be interested in understanding how (if) this modeling approach could be extended to the developmental learning of larger linguistic structures (e.g., constructions or other)?

The theoretical description of the TP model and application to understanding how input distributions can impact developmental trajectories is very well explained. However, I think a little more effort should be spent in explaining how the different graphs/visualizations were created. Additionally, sometimes graphs are not referenced explicitly in the text such as in 3.1 in the last paragraph the authors reference a figure indirectly by simply stating “The second illustration demonstrates this”. I think a bit of clean up in these areas and making sure the graph/visualizations are clearly explained despite space constraints will greatly benefit this paper overall.

Small edits: In 3.1 there is a typo, mainly “If irregulars are distributed uniformly [throughout] the distribution of types…” should replace “If irregulars are distributed uniformly [through out] the distribution of types…”.

---

### Official Review · Reviewer_D2pM · 2022-03-25
**Interesting ideas but scientific contribution is not clear**

**Rating:** 5
**Confidence:** 4

**Review:**

The paper proposes to explain uniformity in children's language learning despite possible differences in individual learning trajectories. The authors investigate these ideas using the Tolerance Principle (TP) (Yang, 2016).

I think the idea is interesting and the use of TP is a good way to approach this question. I do however have several comments/concerns that the authors need to address before they consider a publication.

-First, I am sure if CMLC is a suitable venue since the paper does not use computational linguistic methods

-While TP is a very simple concept, its description in the paper is extremely convoluted and unintuitive. Besides, the authors made use of highly ambiguous word choice and phrasing that impedes understanding for people who are not familiar with TP. To illustrate, the caption in Figure 1 says "If e lies below θ, then the learner should acquire *it* and memorize the exceptions" here the reader would think that *it* refers to "e" whereas it should refer to the "productive rule" (which is not mentioned at all in the caption). The authors also talk about the "sing-sang" pattern, which is highly confusing. The pattern they mean is not sing-sang but "-ing -> -ang". Sing-sang is an example, but the author never makes this clear. I highly recommend the author to rewrite several parts of the paper to make it understandable. As is, the writing is unnecessarily cryptic and does not allow readers to understand what the authors are doing.

-The authors do several simulations that are in my opinion totally unnecessary: the formula is extremely simple, and does not require more clarification through simulations. In fact, these simulations are counter-productive because they are harder to understand than the formula itself!

-In my opinion, the only aspect of this work that is slightly novel is the fact of presenting words with different frequency distribution and showing that this leads to different learning trajectories. However, here again, I am afraid the result is obvious and doesn't require simulations. It is discussed in more intuitive terms in Yang 2016 and subsequent work and the concept of "transient" periods in learning trajectories is talked about lengthily.

Sure, if you do some weird sampling of words (by selecting the lowest frequency, lowest and highest, etc;.) you end up with different transient modes, but these are not insightful about real development since the sampling performed by the author is extreme/artificial and does not correspond to variability in the order of acquisition in children. It would have been more interesting if the authors took real longitudinal vocab acquisition (for example using http://wordbank.stanford.edu/) and showed there to be real variability in transient periods of productive vs. unproductive rule depending on variability in order of word learning of real children.

The part about CHILDES is completely disconnected from everything else. The authors use only one learning trajectory of how vocabulary grows, so we don't see any variability in the learning trajectory. In fact, this part is just duplicating the work in Yang 2016 who used exactly the same phenomena (i.e.,  the -ed rule vs. ing-ang) and the same CHILDES data.

To conclude, while I am sympathetic to the underlying ideas, the work does not make a solid case for it or present anything that we don't already know. In my opinion, the simulations only muddy the water (sorry!) instead of adding insights. Instead of simulations, I really encourage the author to use real data of variability in children learning trajectories and study the extent to which it follows the predictions of TP.

---

### Official Review · Reviewer_rnzi · 2022-03-26
**Review of Modeling the Relationship between Input Distributions and Learning Trajectories with the Tolerance Principle**

**Rating:** 7
**Confidence:** 3

**Review:**

## Overall

This paper shows via computational simulations that the Tolerance Principle (TP; Yang 05), a theory of rule learning in child language acquisition, can explain previously attested uniformity across child learners of English in (a) which morphological patterns are acquired as productive rules and (b) the timecourse of rule acquisition. The TP defines a deterministic threshold over the proportion of exceptions that can be "tolerated" by a productive rule. Under the assumptions of the TP, the authors show empirically that rules may move in and out of productivity during learning due to variation in the relative number of exceptions learned, that the shape of these learning trajectories is influenced by the token frequencies of exceptions, and that simulations using data-driven token frequencies roughly replicate the timecourse of rule learning attested in studies of child language learning.

The writing and interpretation are generally clear, the motivation and methods are sound, and the results support the conclusions. I think this work is appropriate for CMCL, although the breadth of interest may be limited by how embedded it is within a particular theory (Yang 05).


## Major

- The TP framework assumed by this work takes for granted that children tasked with morphological rule learning can already (1) unambiguously segment words in running speech, (2) correctly identify the types of these segmented words as well as any lexical features that could be used to define domains of rules (e.g. syntactic category), and (3) conduct a global search over their entire vocabulary and hypothesized rule set each time a new word is learned, since this is required in order to determine (a) which words fall under the domain of application of each rule and (b) how many of these are exceptional, in order to decide whether to retain the rule according to the tolerance principle. No attempt is made here to show that the TP is more successful than some alternative theory at explaining a set of facts, so the results may be of limited interest to readers who reject one or more of these assumptions.
- About 1/3 of the introduction is dedicated to discussing classical U-shaped trajectories (overregularization) from the acquisition literature, but this work isn't about overregularization. Neither the TP itself nor the computational simulations conducted here concern whether/when a learned rule will be incorrectly applied to an exception. The only question at issue here is when rules are learned in the first place (which certainly could be a necessary condition for overregularization, but not the whole story). Likewise, the abstract concludes that the study shows the TP underlies cross-linguistic differences in learning trajectories, but this again seems like a red herring. This study only uses data from English past tense verb morphology and says nothing directly about cross-linguistic variation. Sure, the result implies that languages with different token frequencies will have different trajectories, but whether this model replicates any known cross-linguistic differences is never evaluated. Thus, the framing of the abstract+intro seems to set up different questions than the ones the authors ultimately address.

## Minor

- The mathematical notation in S3.1 confusingly overloads variable names. In reality, there are 4 quantities involved: (a) the true in-domain vocabulary size, (b) the true number of exceptions, (c) the infant's vocabulary size at a particular point in learning (the sample size parameter of the hypergeometric distribution), and (d) the number of exceptions learned by the infant at a particular point in processing (the support of the hypergeometric distribution). But the authors use $N$ to refer to both (a) and (c) and $e$ to refer to both (b) and (d), so I had to puzzle over that section for a while in order to figure out how the hypergeometric distributions were being parameterized and thus what the plots were representing. Clearer notation would be helpful.
- The plot axes (e and N-e) are a bit of a head trip because they both depend on e. Why not plot e vs N? Barring that, I was able to get my head around the plots by mentally relabeling the y axis as "Num irregular" and the x axis as "Num regular". Revising the axis labels along these lines would clarify things a lot.

---

### Decision · Program_Chairs · 2022-03-29

Accept